# Fine-grained Trace-driven Performance Modeling and Simulation for Large-scale ML Training

Mingyu Liang*, Hiwot Tadese Kassa†, Wenyin Fu†, Brian Coutinho†, Louis Feng†, Christina Delimitrou‡

*Cornell University. *ml2585@cornell.edu*
†Meta. {*hiwotkassa, wenyinfu, bcoutinho, lofe*}*@meta.com*
‡Massachusetts Institute of Technology. *delimitrou@csail.mit.edu*

*Abstract*—**The widespread adoption of machine learning (ML) models, especially with the emergence of large language models (LLMs), has introduced growing challenges in understanding and optimizing both the models and their deployment systems. Performance modeling plays an essential role to predict model performance across various scenarios and guide optimization techniques. In this work, we present TraceSim, a fine-grained, trace-driven performance modeling and simulation framework. TraceSim captures runtime details of models without any model-specific instrumentation, and constructs a comprehensive execution graph to describe model execution. Evaluation with GPT-3 on a production-scale cluster of 256 GPUs achieves, on average, 95.6% accuracy in reproducing the end-to-end execution time, and up to 99.5% accuracy in predicting performance for unseen scaled-up scenarios.**

## I. INTRODUCTION

Over the past few years, ML, notably with the advent of LLMs, has reshaped numerous aspects of daily life. The abundance of available data, combined with advancements in computational resources, has propelled the creation of increasingly intricate models. With the rapid evolution of ML models in both architecture and scale, significant computational resources are now necessary for training. It is crucial to delve into the challenges and opportunities posed by these models and their deployment systems.

Understanding and enhancing ML systems is an increasingly popular domain. Recent studies have demonstrated that employing various techniques for ML system optimization, such as mixed precision [19], pruning [7], operator fusion [5], data/model/tensor parallelism [11], and kernel auto-tuning [3], can yield substantial benefits toward achieving these objectives. However, determining "what" to optimize and "why" optimization is necessary can pose even greater challenges compared to the methods of optimization.

Building an accurate performance model capable of predicting an ML model's performance is fundamental for many optimization studies. While studies like AMPeD [10] employ analytical models to predict performance for the entire forward/backward pass with exposed model parameters, most studies [1], [5], [8], [19] construct low-level execution graphs and then use simulation to predict the end-to-end execution time. The execution graph serves as the foundation of these performance models and its comprehensiveness directly influences their accuracy. However, with the rapid evolution of models and deployment systems, especially in distributed

LLM training, existing approaches have failed to capture and incorporate the new features present in these models, such as concurrent CUDA streams and inter-stream dependencies.

In this work, we present TraceSim, a fine-grained trace-driven performance modeling and simulation framework for large-scale ML training. It currently supports PyTorch models and is adaptable to other ML frameworks. Compared with previous endeavors like Daydream [19] and dPRO [5], our main contributions are:

- TraceSim utilizes the most detailed runtime information about a model, including operators, CUDA runtime events, and kernels. It identifies all possible dependencies among them and constructs a comprehensive execution graph at operator- and kernel-level simultaneously. This fine granularity also allows it to reproduce the detailed execution statistics through simulation, enabling more downstream performance analysis beyond simply predicting the execution time.
- TraceSim only uses the built-in profiling tools from ML frameworks, such as PyTorch Kineto [16]. It does not require any custom instrumentation into the models or frameworks, therefore minimizing the profiling effort.
- TraceSim is evaluated with GPT-3 on a production-scale cluster comprising a total of 256 NVIDIA H100 GPUs, and achieves 95.6% accuracy when predicting the model's end-to-end execution time. Furthermore, we demonstrate a use case of TraceSim for predicting the performance of scaled-up scenarios, achieving up to 99.5% accuracy across various configurations.

## II. RELATED WORK

### A. Profiling Tools and Traces

As the ML system stack rapidly evolves, profiling tools are crucial for understanding the execution characteristics of models and pinpointing performance bottlenecks. With the widespread use of hardware accelerators, such as GPUs and TPUs, hardware vendors also provide associated profiling tools like NVProf [14], CUPTI [12], and Nsight [13]. These tools can expose hardware performance counters, enabling developers to gain valuable insights into the performance characteristics of their models and optimize them effectively.

For better interpretability of the profiling results, ML frameworks also feature built-in tools to collect execution statistics

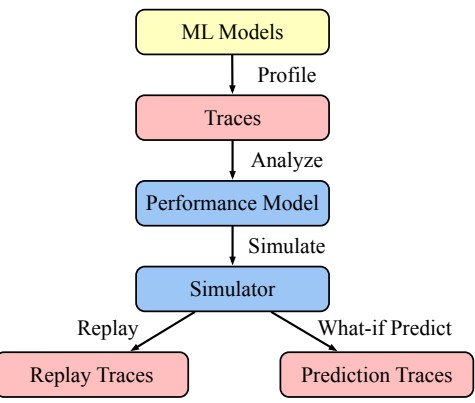

Fig. 1. Overview of TraceSim.

for their framework operators. These tools often integrate with hardware-level traces to provide a comprehensive view of the entire stack, from host to device. For example, PyTorch Kineto [16] utilizes CUPTI [12] to record runtime information for PyTorch operators, CUDA runtime events, and GPU kernels, seamlessly linking them together for a holistic perspective on model execution.

### B. Performance Modeling, Simulation, and Optimization

There has been extensive research utilizing traces to construct performance models, simulate executions, and offer insights for optimization [4]–[6], [8], [15], [18], [19]. Ousterhout et al. use blocked time analysis to understand the I/O, network and stragglers for data analytics applications [15]. ASTRA-sim [18] stands out as a distributed ML system simulator that facilitates the exploration of bottlenecks and the development of efficient methodologies for large DNN models, with a focus on networking aspects.

Daydream [19] predicts model runtime under specific optimizations based on the kernel-level dependency graph collected with CUPTI, while dPRO [5] tracks dependencies among operators and constructs a global data flow graph to estimate DNN model training. However, these approaches do not consider concurrent streams and inter-stream dependencies, leading to significant inaccuracies for today's large-scale distributed training workloads, as we will demonstrate later in our design and evaluation section.

## III. DESIGN

### A. Overview

Figure 1 shows the overview of TraceSim, our trace-driven performance modeling and simulation tool for ML models. First, we collect the profiling traces of the models' execution during runtime. These traces are then forwarded to the analyzer, which extracts essential meta-information required to describe and model the execution, essentially forming a fine-grained execution graph. Finally, the execution graph is fed into the simulator, where we have the option to faithfully simulate the execution using the original trace data or adjust the behavior of certain operators/kernels to emulate execution for what-if scenarios and make performance predictions.

We now focus on PyTorch models as a first step because of its widespread adoption in both industry and academia, as well as the extensive profiling capabilities it offers. However, our approach can be extended to support other ML frameworks, such as TensorFlow and MXNet.

### B. Profiling Traces

We collect profiling data using PyTorch Kineto [16], which provides details for all PyTorch operators, CUDA runtime events and GPU kernels, including name, start time, duration, CUDA stream ID, thread ID, and more. In contrast to previous approaches like Daydream which requires framework and model instrumentation, our profiling only requires a few lines of codes, improving the ease of use with minimal effort.

For additional use cases, such as the scaled-up prediction discussed in Section IV-C, we leverage Chakra [2] as a complement to Kineto. Chakra is officially supported by PyTorch and can be seamlessly collected alongside Kineto. We will delve into the specifics of why and how we use Chakra later.

### C. Execution Graph

The essence of a model's execution lies in its execution graph, which is depicted by the tasks being executed and the relationships between them.

*1) Tasks:* Our execution graph contains the following two types of tasks:

**CPU tasks**: We classify all tasks executed on the CPU, including PyTorch operators and CUDA runtime events, as CPU tasks. We record the duration of each CPU task along with the CPU thread on which it is executed.

**GPU tasks**: We classify all tasks executed on the GPU as GPU tasks, which primarily consist of GPU kernels. We record the duration of each GPU task along with the CUDA stream on which it is executed.

*2) Dependency:* We identify four types of dependencies that encompass all possible relationships between tasks.

**CPU to CPU**: This dependency includes both the intra-/inter-thread dependency between CPU tasks. CPU tasks assigned to the same thread naturally execute in a serialized manner. Therefore, we add a dependency between each pair of consecutive CPU tasks within the same thread as intra-thread dependency. Regarding inter-thread dependency, there are cases where CPU tasks from one thread are blocked by tasks from another thread. For example, in PyTorch, a dedicated thread handles the backward pass, and its first operator needs to wait for the last operator on the forward thread to complete. We identify such dependencies by detecting significant gaps in execution within each thread and build dependencies across threads accordingly.

**CPU to GPU**: A GPU task is typically initiated by a corresponding CPU-sided CUDA event, such as cudaLaunchKernel. In Kineto trace, each CUDA runtime event and GPU kernel is associated with a correlation ID. We utilize this ID to add the dependency between CPU tasks and GPU tasks.

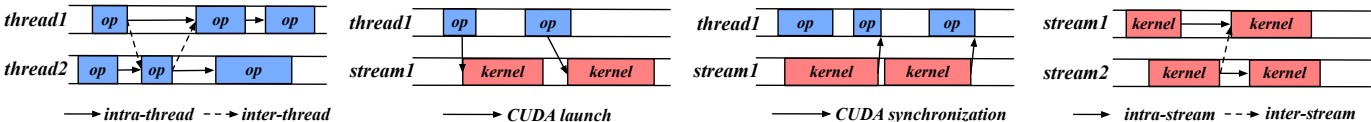

Fig. 2. Four types of dependencies between the tasks.

**GPU to CPU**: CUDA synchronizations are frequently encountered during model execution. When a CUDA synchronization event, such as cudaDeviceSynchronize or cudaEventSynchronize, is invoked on the CPU, it blocks until all relevant GPU kernels have completed execution. Consequently, this introduces a dependency from one or multiple GPU tasks to a CPU task.

**GPU to GPU**: Similar to CPU to CPU, this dependency includes the intra- and inter-stream dependency between GPU tasks. GPU kernels belonging to the same stream are executed sequentially, thus two consecutive GPU tasks in the same stream have a dependency between. To identify inter-stream dependencies, we leverage the special CUDA synchronization events captured in the Kineto trace. Specifically, we focus on a pair of CUDA events: cudaEventRecord() and cudaStreamWaitEvent(). The kernel launched prior to cudaEventRecord() serves as the source kernel, while cudaStreamWaitEvent() specifies a stream, and the next kernel scheduled on that stream becomes the dependent kernel. By analyzing these events, we can effectively detect dependencies between GPU tasks across different streams.

This execution graph is similar to the approach proposed in Daydream, as we find it necessary and effective to build a performance model by constructing the dependencies among low-level tasks. However, we have implemented several improvements. First, our profiling tool can capture detailed information about communication operators/kernels, eliminating the need for manual insertion of communication tasks via model instrumentation and estimation of performance. Second, Daydream does not support concurrent streams and serializes all GPU tasks' execution on a single stream, and consequently it does not consider any inter-stream dependency. However, overlapping the execution of kernels is now a very common practice to hide the communication overhead in distributed training. Whether two kernels can execute in parallel is determined by available stream resources and kernel-level dependencies, which directly influences the end-to-end execution time and other execution characteristics. This limitation is also present in dPRO, which utilizes only operator-level traces. In our evaluation section, we demonstrate the importance of considering these factors, and showcase the improvements our approach offers compared to existing work.

### D. Simulation

We utilize this execution graph as input for our event-based simulator written in Python. CPU and GPU tasks are assigned to their respective threads and streams, and executed according to their dependencies. The simulator will generate a new Kineto trace that includes comprehensive details of all tasks.

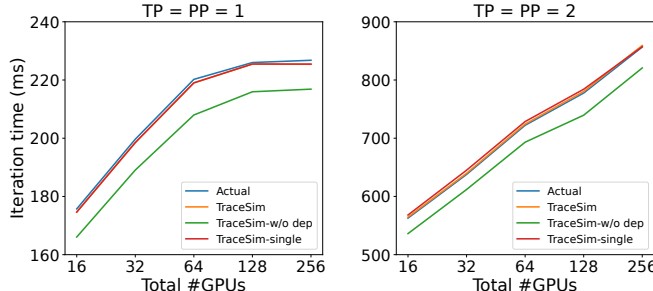

Fig. 3. Simulated execution time per iteration.

This output trace can serve as a simulated execution, enabling further analysis and evaluation of performance across different conditions and scenarios. Additionally, the simulation framework allows manipulation of task execution to emulate various what-if scenarios, as we will demonstrate in Section IV-C.

## IV. EVALUATION

### A. Methodology

We conduct our evaluation of TraceSim on a cluster of 32 servers equipped with a total of 256 NVIDIA H100 GPUs. We use the GPT-3 model [9] open sourced in MLPerf and shrink its size to 125M to fit in our smallest setup in terms of GPUs. We collect the traces and evaluate the performance under various parallel strategies, including TP (Tensor Parallelism), PP (Pipeline Parallelism) and DP (Data Parallelism). We use CUDA 12.0 and PyTorch 2.2 as our testing environment.

The main contribution of TraceSim compared to existing work is the ability to handle concurrent streams and inter-stream dependencies. To demonstrate the importance of that, we implement two baseline versions of TraceSim for comparison: one is **TraceSim-single**, which does not consider concurrent streams, with only one stream for computation and one for communication. The other is **TraceSim-w/o dep**, which does not account for inter-stream dependencies.

### B. Same-scale simulation

*1) End-to-end execution time:* Figure 3 shows the simulated per-iteration execution time under various settings. Notably, TraceSim accurately reproduces the execution time, closely aligning with the actual time with an average accuracy of 95.6%. This demonstrates that our fine-grained execution graph can effectively capture the essential information of the model's execution. Comparatively, the execution time simulated by TraceSim-w/o dep consistently underestimates the actual results. This is attributed to the lack of consideration for inter-stream dependencies, leading to additional parallel execution that does not exist in the actual execution. In contrast,

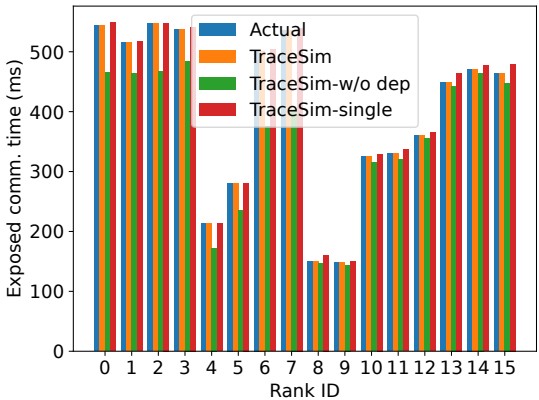

Fig. 4. Exposed communication time per rank.

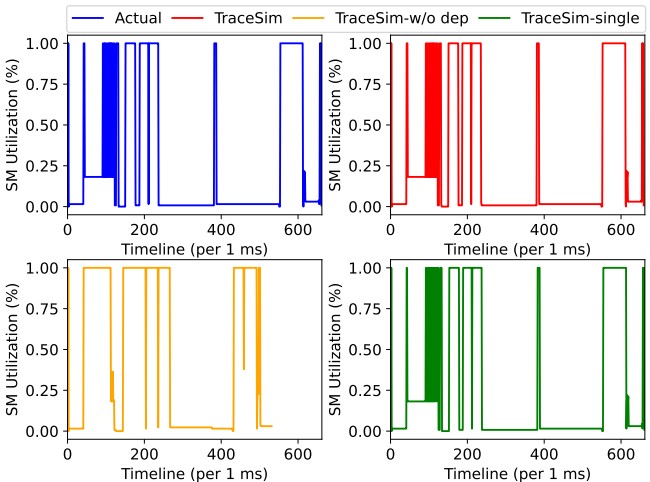

Fig. 5. SM utilization of one rank.

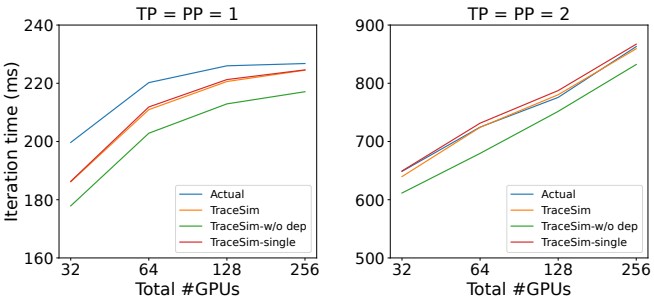

Fig. 6. Predicted execution time per iteration.

TraceSim-single consistently overestimates the execution time since it prohibits any parallel execution. Although considering concurrent streams may not exhibit significant differences in our current results, it is crucial for enabling the overlap of more kernels. While our tested model features multiple communication streams, limited overlap occurs between kernels on these streams due to inter-stream dependencies. We expect a greater impact in other models where more overlap occurs.

*2) Exposed communication time:* Since the simulation can emulate the model's execution with per task details, including start time, duration, and on which thread/stream a task is executed, we can conduct more downstream performance analysis beyond simulating the end-to-end execution time. In large-scale distributed training, such as with LLMs, a significant portion of time is dedicated to communication and synchronization events between GPUs. Engineers are striving to maximize overlap between the execution of computation and communication kernels. One important metric they examine is the exposed communication time, defined as the time spent in communication that does not overlap with computation.

Figure 4 presents the exposed communication time for different ranks, based on traces collected with TP = PP = 2 and DP = 4. Compared with the baselines, TraceSim achieves highest accuracy compared to the actual model, demonstrating that our fine-grained execution graph is necessary to accurately reproduce the execution details.

*3) SM utilization:* In Figure 5, we showed Streaming Multiprocessor (SM) utilization of one rank collected with TP = PP = 2 and DP = 4. The utilization is derived from the trace by analyzing the number of SMs used by each kernel and aggregated per 1ms timestep. Similarly to the exposed communication time, it again proves the necessity of considering concurrent streams and inter-stream dependencies to accurately reproduce the detailed execution statistics.

### C. Scaled-up Prediction

Debugging and evaluating the performance of large-scale ML training pose significant challenges due to the high cost and complexity associated with the testing setups. There is a need for a tool capable of predicting a model's large-scale performance using more manageable setups. In this section, we demonstrate a use case of TraceSim to predict the scaled-up performance of models with small-scale traces, such as using traces collected with 16 GPUs to predict the performance of 128, 256, or even more GPUs.

We start with a simple scenario that predicts the scaled-up behavior of models by changing the DP only, since this will not change the computation of each rank, so that we only need to alter the communication part. We utilize the execution graph constructed from the 16 GPUs traces as input for the simulator. During the simulation, we manipulate only the behavior of communication operators/kernels by substituting their execution time with pre-collected data for the target larger scales. We assume we can obtain the execution time of various communication operators at larger scales in advance. Currently, we collect the actual traces of large scales as ground truth, enabling us to directly analyze these traces and extract this execution information. Since the behavior of communication operators is only determined by the arguments (e.g., data size, communicator group) and network environment, this information only needs to be collected once and can be easily reused for various models and simulation purposes. Additionally, this information can also be simulated using network simulators like ASTRA-sim [18] or predicted using analytical models [10], [17]. As Kineto trace lacks arguments information, we collect a Chakra trace [2] to supplement it.

Both traces can be collected simultaneously for the same iteration, allowing us to easily link them together.

Figure 6 illustrates that we can accurately predict the per-iteration execution time of larger scales using the traces collected with 16 GPUs. When scaling only DP, the average accuracy of execution time prediction is 97% with TP = PP = 1 and 99.5% with TP = PP = 2. This prediction capability holds significant potential in eliminating the need for expensive hardware resources when conducting performance debugging or fine-tuning tasks. By reusing the execution graph built from small-scale traces and altering only the necessary parts, we can accelerate the iterative process, making performance evaluation and optimization more efficient and cost-effective.

## V. Conclusion

We present TraceSim, a trace-driven performance modeling and simulation framework for large-scale ML training. By constructing a fine-grained execution graph, TraceSim is able to capture and reproduce a model's execution. Its fine granularity and flexibility also opens up many interesting use cases, such as scaled-up performance prediction.

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
