# OpenReview forum: "Fine-grained Trace-driven Performance Modeling and Simulation for Large-scale ML Training"
_iscaconf.org/ISCA/2024/Workshop/MLArchSys — MLArchSys 2024 OralPoster_

### Official Review · Reviewer_BfPM · 2024-05-23
**Useful tool but the idea is not entirely new. Evaluation can be improved.**

**Confidence:** 4
**Rating:** 5

**Detailed Feedback And Questions For Authors:**

The paper presents a useful tool for ML model’s performance estimation via trace simulation. The tool doesn’t require any custom instrumentation to the model. The method is sound, but not entirely a new concept. A similar method of profiling computation ops on hardware to estimate the model’s total runtime is regularly used in many ML systems papers. However, the proposed method is more flexible as it can handle concurrent execution.  This trace based approach also has an advantage over a static approach that estimates performance from a given computation graph (without execution trace). In particular, the static approach may incur significant inaccuracy in a presence of dynamic loop bound (e.g. dynamic sequence length). I wish that the paper compares against this static approach.


The usefulness of this tool is to help find optimal configurations of a model. There is a wide range of configurations that one can optimize for, e.g. parallelism, fusion, rematerialization / checkpointing strategies, etc. However, the paper evaluates the accuracy of the method on only one ML model (GPT-3) on only two parallelism strategies (TP = PP = 1 and TP = PP = 2, with varying numbers of GPUs). It would be good to evaluate the method on diverse types of models and a more diverse space of parallelism strategies (e.g. from Megatron, Alpa, PipeDream, FlexFlow) and other types of configurations. Additionally, assume that we use the proposed method to help find the best parallelism strategy on a large number of GPUs. While we can collect a trace of each candidate on a small number of GPUs, if a parallelism strategy requires a different communication pattern, then the proposed method still requires profiling the runtime of the communication ops on a large number of GPUs. The paper should discuss how to effectively support this sort of exploration.

It seems like the method may infer unnecessary intra dependencies. For example, it is possible that op B runs after op A on a GPU without actually depending on op A, but the tool will infer that B depends on A. This may hamper estimation accuracy in some models. Regarding the concurrency (multiple GPU streams) support, the current set of experiments do not effectively show the importance of this feature. Perhaps, evaluating more classes of models will highlight the effect better.

**Top Reasons To Accept The Paper:**

* A potentially useful method and tool for accurate performance estimation
* Doesn’t require any user customization
* More accurate than static approach

**Top Reasons To Reject The Paper:**

* Not entirely a new idea in ML performance domain
* Evaluation on only one model and one what-if analysis with very few and fairly simple parallelism strategies

---

### Official Review · Reviewer_vi7m · 2024-05-24
**This paper presents TraceSim, a fine-grained, trace-driven performance modeling and simulation framework for optimizing LLMs, demonstrating advancements in prediction accuracy and profiling efficiency, but lacks detailed analysis of communication tasks in distributed training, extensive comparisons with similar approaches, and discussions on overheads and generality.**

**Confidence:** 4
**Rating:** 6

**Detailed Feedback And Questions For Authors:**

The paper presents TraceSim, a fine-grained, trace-driven performance modeling and simulation framework designed to address the growing challenges in understanding and optimizing machine learning (ML) models, particularly large language models (LLMs). Key challenges targeted by this paper include the accurate prediction of model performance across diverse scenarios and the need for optimization without requiring extensive model-specific instrumentation. TraceSim constructs a comprehensive execution graph by capturing detailed runtime information, such as operators, CUDA runtime events, and kernels, and identifies dependencies among them. Unlike previous approaches, TraceSim relies solely on built-in profiling tools from ML frameworks, minimizing the profiling effort. The framework's evaluation with GPT-3 on a 256-GPU cluster demonstrates high accuracy in reproducing end-to-end execution times and predicting performance for scaled-up scenarios.

- It could have been beneficial if the authors had commented on modeling the TP or PP greater than one (e.g., Figure 3 right). Also, in Figure 3 (left, TP=PP=1), if the orange line (TraceSim) completely overlaps with the blue line (Actual), it could be better to visualize it in a way that illustrates this overlap clearly.

- The evaluations of this paper can be further improved by including comparisons with prior studies (especially those that use similar approaches such as Daydream) and expanding the analysis to cover various aspects, pros and cons, and trade-offs.

- As mentioned in the paper, communication tasks play a crucial role in large-scale distributed training. The paper could have included more details about this aspect compared to prior studies. For instance, how does TraceSim capture detailed information about communication operators/kernels? Is the elimination of manual insertion of communication tasks a novel aspect of this work? Additionally, when including the communication aspect in distributed training, how do the authors compare TraceSim with other recent studies such as Calculon [SC’23]?

- Discussions about the overheads of running TraceSim and its generality to other models and system setups would be very helpful.

**Top Reasons To Accept The Paper:**

TraceSim provides a comprehensive and highly accurate performance modeling framework for large-scale ML training, demonstrating significant advancements in prediction accuracy and reducing profiling effort by utilizing built-in tools, which are crucial for optimizing LLM deployments.

**Top Reasons To Reject The Paper:**

The paper lacks detailed analysis of communication tasks in distributed training, does not compare extensively with similar approaches, and omits discussions on overheads and generality, limiting its comprehensiveness and practical applicability in broader contexts.

---

### Official Review · Reviewer_fDZo · 2024-05-28
**Review of Fine-grained Trace-driven Performance Modeling and Simulation for Large-scale ML Training**

**Confidence:** 4
**Rating:** 6

**Detailed Feedback And Questions For Authors:**

Please see the comments in the weakness/areas of improvement section.

**Note: Pasting the same thing from above if authors are not able to see the comments above**

**Summary**
The authors propose TraceSim, a fine-grain trace-based performance modeling framework that uses profilers such as Pytorch Kineto to augment the execution graph with runtimes, including exposed communication. The authors utilize these traces to predict the execution time, exposed communication, SM utilization, and scale-out studies with approximately 95%+ accuracy.

**Strengths**
* *Useful Trace Collection:* The collection of traces and datasets during the training of large language models is particularly beneficial.
* *Native Hardware Profilers* Using native hardware profilers without the need to instrument the code significantly reduces the overhead associated with manual instrumentation.

**Weaknesses/Areas for Improvement**

* *Overhead Cost Discussion*: It would be beneficial if the authors discuss the overhead cost of using profilers for collecting statistics data at the operator level, especially when using CUPTI [1] .
* *Scale-out Study Assumptions:* The scale-out study prediction assumes that the communication costs are available in advance. If such a large scale is available initially, it raises the question of why not use TraceSim to collect the actual trace and then use it to predict for scale-up scenarios. This ties into the previous comment about the overhead in collecting traces for large-scale setups. Addressing these points in the presentation or final version would be helpful.

**Conclusion**
The paper has significant potential in creating traces and datasets, which is beneficial for the research community. These datasets have potential applications beyond what the authors cover in this work. The authors might also be considering using machine learning [2] to predict the cost of runtime, exposed communication, and future scale-out studies . For these reasons, I am inclined to accept the paper. It would be beneficial if the authors address the weaknesses and areas for improvement in their final version or during the workshop presentation.

**References**

[1]: RL-Scope: https://arxiv.org/pdf/2102.04285

[2]: TPUGraph: https://arxiv.org/abs/2308.13490

**Top Reasons To Accept The Paper:**

**Summary**
The authors propose TraceSim, a fine-grain trace-based performance modeling framework that uses profilers such as Pytorch Kineto to augment the execution graph with runtimes, including exposed communication. The authors utilize these traces to predict the execution time, exposed communication, SM utilization, and scale-out studies with approximately 95%+ accuracy.

**Strengths**
* *Useful Trace Collection:* The collection of traces and datasets during the training of large language models is particularly beneficial.
* *Native Hardware Profilers* Using native hardware profilers without the need to instrument the code significantly reduces the overhead associated with manual instrumentation.

**Weaknesses/Areas for Improvement**

* *Overhead Cost Discussion*: It would be beneficial if the authors discuss the overhead cost of using profilers for collecting statistics data at the operator level, especially when using CUPTI [1] .
* *Scale-out Study Assumptions:* The scale-out study prediction assumes that the communication costs are available in advance. If such a large scale is available initially, it raises the question of why not use TraceSim to collect the actual trace and then use it to predict for scale-up scenarios. This ties into the previous comment about the overhead in collecting traces for large-scale setups. Addressing these points in the presentation or final version would be helpful.

**Conclusion**
The paper has significant potential in creating traces and datasets, which is beneficial for the research community. These datasets have potential applications beyond what the authors cover in this work. The authors might also be considering using machine learning [2] to predict the cost of runtime, exposed communication, and future scale-out studies . For these reasons, I am inclined to accept the paper. It would be beneficial if the authors address the weaknesses and areas for improvement in their final version or during the workshop presentation.

**References**

[1]: RL-Scope: https://arxiv.org/pdf/2102.04285

[2]: TPUGraph: https://arxiv.org/abs/2308.13490

**Top Reasons To Reject The Paper:**

I think the paper is decent and trending in the right direction. The paper can certainly be improved and encourage the authors to address the comments in the weakness/areas of improvements section.

---

### Official Review · Reviewer_XSWk · 2024-05-28
**Light weight simulation for ML training but lacks comparison.**

**Confidence:** 3
**Rating:** 5

**Detailed Feedback And Questions For Authors:**

The proposed techniques show significantly good accuracy in predicting execution time for production-scale systems. The techniques are well discussed but more work could be done on the motivation section. The paper writing needs some reorganizing. While the comparison against the real world shows good prediction accuracy, overall evaluation is limited. Please address the comments below.

1. Figure 2 is not well-referenced or discussed in the paper.

2. More discussion is required for the overall methodology as shown in Figure 1. What kind of analysis can be performed with a trace generated once? Analysis of what components would require profiling again?

3. What kind of “what if analysis” can be done or not done with the traces? Any insights that can be obtained?

4. For evaluation, comparison with state-of-the-art work i.e., Daydream and dPRO are missing. What are the trade-offs for existing related works for simulating LLM training?

5. What is the generality of the proposed method over different LLM models? Any challenges in modeling different LLM models?

6. In Figure 6, the simulated execution time for TP=PP=1 seems to have a higher prediction error than TP=PP=2. Any specific reason for that?

**Top Reasons To Accept The Paper:**

The paper proposes lightweight and efficient trace-driven simulation for LLM training that does not require much instrumentation. The evaluation shows that the proposed technique achieves high accuracy in predicting the performance of unseen configurations.

**Top Reasons To Reject The Paper:**

The paper lacks evaluation and comparison against existing state-of-the-art works like DayDream and dPRO.

---

### Official Review · Reviewer_eqtk · 2024-05-28
**The paper presents TraceSim, a fine-grained, trace-driven performance modeling and simulation framework. But the paper does not clarify additional value added by the framework over existing techniques and lacks advancement of the field..**

**Confidence:** 2
**Rating:** 3

**Detailed Feedback And Questions For Authors:**

I am missing the key contribution or the key problem that is being solved by the framework which is not yet achieved using existing frameworks.

**Top Reasons To Accept The Paper:**

None.

**Top Reasons To Reject The Paper:**

It is unclear what is main contribution in the paper and key differences to existing framework like AstraSim. i.e. I am missing what is the key contribution and advancement by the trace driven performance modeling.

---

### Decision · Program_Chairs · 2024-05-30

**Decision:**

Accept (Oral/Poster)

**Comment:**

Congratulations! We are pleased to inform you that your paper has been accepted for presentation at MLArchSys 2024. We look forward to your participation at the workshop. Further details regarding the schedule and format will be provided soon. See you at the workshop!